# Decline in HIV prevalence among female sex workers in Zimbabwe between 2013 and 2023

Sungai T. Chabata [1,2], Harriet S. Jones [3], Tsitsi Hove [1], M. Sanni Ali[4], Loveleen Bansi-Matharu[5], Fortunate Machingura[1,2], Albert Takaruza[1], Primrose Matambanadzo[1], Jeffrey Dirawo[1], Richard Steen[2], Joanna Busza [3], Raymond Yekeye[6], Owen Mugurungi[7], Andrew N. Phillips [5], Frances M. Cowan [1,2] ✉ & James R. Hargreaves[3]

HIV epidemic trends among female sex workers in sub-Saharan Africa are rarely known. We analysed HIV prevalence trends among 10,562 female sex workers aged 18–39 years, recruited through serial cross-sectional respondent-driven sampling surveys in 13 towns and 2 cities in Zimbabwe between 2013 and 2023. HIV prevalence remained stable from 2013 to 2016–2017 but declined significantly from 54.6% in 2016–2017 to 38.9% in 2021–2023 (cluster prevalence mean difference: 15.7%, prevalence ratio: 0.74; 95% CI: 0.68-0.79). This decline cannot be attributed to sampling bias or shifts in the characteristics of the female sex worker population. Mathematical modelling using the HIV Synthesis model and age cohort analysis also suggested lower HIV incidence in later years. While the availability of oral pre-exposure prophylaxis increased, there was little evidence of reduced risk behaviour over time. Increased treatment coverage among the male population likely contributed to the lower HIV incidence among female sex workers.

As in many countries in sub-Saharan Africa, HIV prevalence in Zimbabwe has declined among adult women since the epidemic's peak, falling from 26.5% in 1997 to 13.2% in 2016 and 11.8% by 2020[1,2]. HIV incidence has also declined[3]. Female sex workers (FSW) are at high risk of HIV compared to women in the general population[4], but epidemiological trends among this group are rarely described[5]. The importance of FSW for population level HIV transmission in Africa has been growing[6]. As population HIV incidence declines, understanding trends in infections among populations at high risk of HIV infection becomes increasingly important[7].

In a recent meta-analysis, HIV incidence among women who engage in sex work was estimated to be 4.9 times higher than among women in the general population in Eastern and Southern Africa[3]. However, longitudinal studies among FSW are rare, and changes over time remain poorly understood[3]. The challenges in conducting representative longitudinal studies to estimate HIV incidence among FSW stem from the impracticalities of obtaining a population sampling frame due to the hidden and transitional nature of sex work. Furthermore, where FSW have been followed over time in cohort studies, these have reported declines in HIV incidence associated with changes in behaviours as a result of cohort participation, i.e. Hawthorne effects[8–10]. High levels of loss to follow up among mobile and transient populations such as FSW[11] also make cohort studies difficult to conduct and interpret[12,13]. Continued stigma and prohibitive legal

[1]Centre for Sexual Health and HIV/AIDS Research (CeSHHAR) Zimbabwe, Harare, Zimbabwe. [2]Department of International Public Health, Liverpool School of Tropical Medicine, Liverpool, United Kingdom. [3]Department of Public Health, Environments and Society, London School of Hygiene and Tropical Medicine, London, United Kingdom. [4]Department of Disease Control, London School of Hygiene and Tropical Medicine, London, United Kingdom. [5]Institute for Global Health, University College London, London, United Kingdom. [6]National AIDS Council, Harare, Zimbabwe. [7]AIDS and TB Directorate, Ministry of Health and Child Care, Harare, Zimbabwe. ✉e-mail: Frances.Cowan@lstmed.ac.uk

environments pose further challenges[14]. Systematic reviews report that heterogeneity in sampling approaches, inclusion criteria and definitions of sex work make survey findings difficult to compare over time[3].

In 2009, we established the Sisters with a Voice (Sisters), now known as the Key Populations (KP) programme, in Zimbabwe on behalf of the Ministry of Health and Child Care (MoHCC) and the National AIDS Council (NAC)[6,15–20]. By 2023, the programme had seen over 164,000 women, providing clinical services for sex workers nationally within 154 primary care clinics, in line with WHO guidelines (see Methods, Study setting for more details). Since its inception, the programme has hosted a unique implementation science research platform nested within service delivery[6,16,21–24]. We have conducted multiple respondent-driven sampling surveys (RDS) in programme sites using a consistent method over more than 10 years. RDS, by using associated statistical adjustments to account for recruitment biases, is a robust and practical approach to probability-based sampling among hidden populations[25]. We have previously reported stable HIV prevalence and evidence of high HIV incidence among FSW in RDS surveys conducted between 2011 and 2016[6]. We have also shown large improvements in the HIV treatment cascade associated with programme expansion between 2013 and 2016[6,23]. Here, we use data from serial RDS surveys to analyse and explain changes over time in HIV prevalence among FSW aged 18–39 across Zimbabwe between 2013 and 2023. These changes may also reflect shifts in HIV incidence, as prevalence is more stable and easier to measure, especially in cross-sectional surveys. We focus our analysis on women under 40 years of age because trends in HIV prevalence in this group are more likely to reflect HIV incidence trends than in older age groups.

We assessed the strength of evidence that HIV prevalence was lower among Zimbabwean FSW in 2021–2023 compared to previous years (2013, 2016–2017) and aimed to establish if this indicated declines in new HIV infections. We therefore explored how HIV prevalence varied by age within surveys, and varied between surveys within age cohorts, as these patterns are associated with HIV incidence. We then investigated possible explanations for the HIV prevalence decline that we observed: specifically, whether the changes reflected: (i) evidence of time-dependent sampling bias across our studies, (ii) changes in the characteristics of women engaged in sex work, (iii) reduced risk-behaviour or improved prevention coverage and (iv) increased engagement in programme services. We triangulated our empirical findings with estimates of HIV prevalence and incidence from a well-calibrated model of the HIV epidemic in Zimbabwe.

## Results

We recruited 15,562 FSW through RDS to five separate multi-site surveys conducted in Zimbabwe in 2013, 2016, 2017, 2021 and 2023. In this analysis we include 10,562 survey respondents, after excluding those from programme sites without repeat surveys across multiple years ($n = 2421$), those over the age of 39 ($n = 2528$), those with missing

RDS recruiter information ($n = 15$) and those with missing HIV test ($n = 36$) (Supplementary Appendix A, Table S1). We analysed data for 6240 FSW aged 18–39 recruited from the same 13 locations in 2013, 2016 and 2021, and for 4322 FSW recruited from the same two cities in 2017 and 2023. To test whether HIV prevalence had declined between 2016–2017 and 2021–2023, we pooled and calculated cluster prevalence mean using data from all 15 sites for 2016–2017 ($n = 4005$) and 2021–2023 ($n = 4475$), excluding data from 2013 ($n = 2082$). Across all five surveys, 5631 FSW tested HIV-negative and 4931 tested HIV-positive.

### HIV prevalence from RDS surveys

Cluster mean RDS-weighted HIV prevalence was 55.1% (cluster range: 40.7–74.6) in 13 programme sites in 2013 and similar in the same sites in 2016 at 55.5% (cluster range: 36.2–72.3), but declined to 39.5% (cluster range: 22.6–50.9) in 2021 (Supplementary Appendix A, Table S2). HIV prevalence also declined in the two cities from 49.4% in 2017 (cluster range: 47.1–50.6) to 33.7% in 2023 (cluster range: 28.4–36.4) (Supplementary Appendix A, Table S2). In the 15-site pooled analysis, cluster mean RDS-weighted HIV prevalence in 2021–2023 was 38.9% (cluster range: 22.6–50.9), which was a 15.7% drop from 54.6% (cluster range: 36.2–72.3) in 2016–2017 (95% CI: 9.2–22.2; $p$-value < 0.001) (Table 1) – relative risk = 0.74 (95% CI: 0.68–0.79) (Supplementary Appendix A, Table S9).

In all survey rounds, HIV prevalence increased sharply with age, starting with a cluster mean HIV prevalence among 18 years old ranging from 3.9% (cluster range: 3.4–4.4) to 27.7% (cluster range: 2.5–52.9) between surveys and reaching between 56.1% (cluster range: 33.8–78.4) and 90.8% (range 88.0–93.6) among 39 year olds (Fig. 1a-c & Supplementary Appendix A, Table S4). Cluster mean age-specific HIV prevalence was between 4.0% (95% CI: 10.2–18.1) and 26.7% (95% CI: 14.5–38.9) lower at all ages in 2021–2023 compared to 2016–2017 (except among 20 year olds where HIV prevalence was 16.9% in both time periods) (Fig. 1a–c & Supplementary Appendix A, Table S4).

Adapting a "synthetic cohort" approach developed by Hallett et al.[26–28], we explored how HIV prevalence among age cohorts had changed between surveys, for example comparing HIV prevalence among 21 year olds in 2016 with 26 year olds 5 years later, in 2021 (Fig. 1d–f & Supplementary Appendix A, Table S5). In the 13 sites with data from 3 survey rounds, HIV prevalence increased by a cluster mean of 4.3% within age cohorts of FSW aged 18–36 in 2013 and aged 21–39 in 2016. However, it declined by a cluster mean of 1.8% within age cohorts of FSW aged 18–34 in 2016 and aged 23–39 in 2021. HIV prevalence declined in most (10/17) age cohorts in the 2016 to 2021 period in the 13 locations, but only in 7 of the 16 age cohorts in the two cities from 2017 to 2023 (cluster mean decline 0.4%). While it is difficult to translate these age cohort prevalence changes into estimates of HIV incidence among FSW, as in the original Hallet et al. paper, the prevalence decline from 2016 to 2021, compared to the increase from 2013 to 2016, implies lower HIV incidence over the later period. This is

**Table 1 | Empirical and modelled HIV prevalence estimates for FSW in Zimbabwe**

| HIV prevalence (RDS data) | Pooled surveys 2016–2017 ($N = 4005$) | | Pooled surveys 2021–2023 ($N = 4475$) | | Difference in means (95% CI) | $p$-value |
|---|---|---|---|---|---|---|
| | $n/N$ | Cluster mean [range]ᵃ | $n/N$ | Cluster mean [range]ᵃ | | |
| HIV prevalence – unweighted (%) | 2126/4005 | 53.1 (33.5 – 71.1) | 1634/4475 | 39.6 (22.2 – 50.3) | –14.5 (–20.9––8.1)) | <0.001 |
| HIV prevalence – weighted (%) | 2126/4005 | 54.6 (36.2–72.3) | 1634/4475 | 38.9 (22.6–50.9) | –15.7 (–22.2––9.2) | <0.001 |
| **Modelled estimates** | **2016** | **Mean (90% range)** | **2021** | **Mean (90% range)** | | |
| HIV prevalence – modelledᵇ (%) | | 58.0 (36.8, 82.9) | | 0.5 (26.5, 76.9) | | |

Source data are provided as a Source Data file. Data are number of participants. P-values (two-sided) are based on t test.
ᵃMean of 15 site specific HIV prevalence estimates with minimum and maximum site estimates.
ᵇ2023 prevalence mean and 90% range: 48.0 (23.7–74.6).

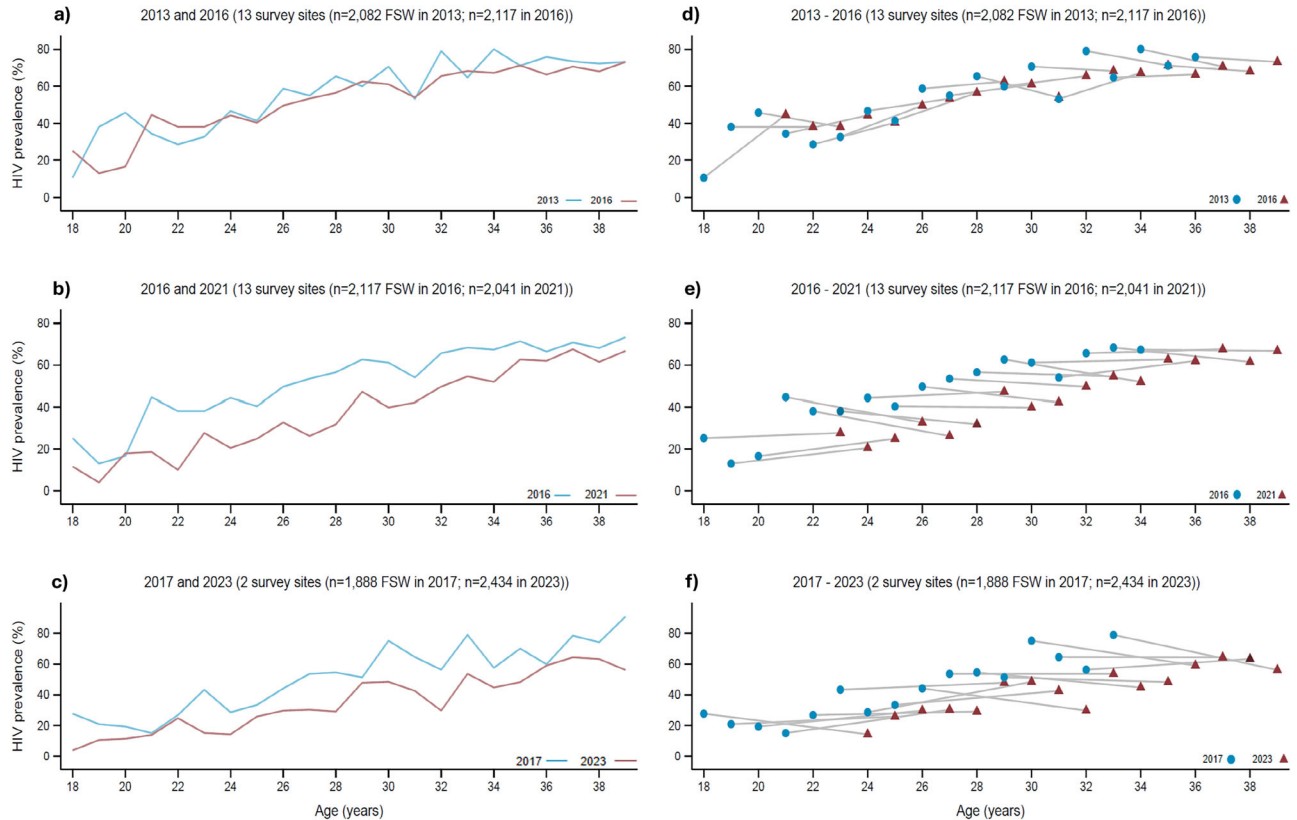

**Fig. 1 | Age-specific HIV prevalence within surveys and age-cohort prevalence between surveys. a–c** Age-specific HIV prevalence within surveys. **d–f** Age cohort prevalence between surveys. **a** age-specific HIV prevalence in 2013 and 2016 surveys; **b** age-specific HIV prevalence in 2016 and 2021 surveys; **c** age-specific HIV prevalence in 2017 and 2023 surveys; **d** HIV prevalence within 2013–2016 age cohorts; **e** HIV prevalence within 2016–2021 age cohorts; **f** HIV prevalence within 2017–2023 age cohorts. Source data are provided as a Source Data file.

especially true given the increases in ART coverage (63.3%, *n* = 751/1159 vs 84.0%, *n* = 669/827) and the proportion of virally suppressed individuals (62.2%, *n* = 757/1159 vs 90.9%, *n* = 747/827) among FSW living with HIV from 2016 to 2021 (Supplementary Appendix A, Table S10), which positively impacts survival rates. The main exception to this pattern was among young FSW in 13 of the sites where HIV prevalence increased by 7.5% among 24 year olds in 2021 compared to 19 year olds in 2016, and by 8.3% among 25 year olds in 2021 compared to 20 year olds in 2016.

**Explaining declining HIV prevalence across RDS survey data**
We investigated potential explanations for the significant decline in HIV prevalence in our pooled data across all 15 sites between 2016–2017 and 2021–2023. First, we examined whether there was evidence of differential sampling bias across the surveys (Table 2). In 2016–2017 HIV prevalence increased from 54.1% to 62.1% between wave 0 and wave 1, and then declined, while in 2021–2023 HIV prevalence consistently declined over the waves of RDS recruitment (Table 2 & Fig. 2). Recruitment homophily with respect to HIV status, representing whether women who tested HIV positive were more likely to recruit other women who tested HIV positive, was low to moderate and similar between 2016–2017 and 2021–2023, at 1.04 (cluster range: 0.91–1.22) and 1.07 (cluster range: 0.92–1.16), respectively. The proportion of seeds with chains reaching the final wave fell by a cluster mean of 21.5% (95% CI: 11.5–31.5; *p*-value < 0.001) in 2021–2023 to 69.2% (cluster range: 33.3–100.0). Comparing the characteristics of the recruited sample with the self-reported characteristics of each respondent's immediate contacts/social networks (egonets) suggested that the sampling procedure may have under-represented both FSW that were younger and older than the median age group, or that FSW

overestimated the extent to which the FSW population are like them. However, this pattern was also similar between surveys. We concluded there was little evidence that sampling bias could explain the HIV prevalence decline observed.

We examined the sociodemographic profile of women engaged in sex work in the 2021–2023 surveys compared to previous surveys (Table 3). The cluster mean age of FSW in our analysis was similar between 2016–2017 and 2021–2023. The women recruited in 2021–2023 were more likely to have completed secondary education (cluster mean difference 10.2% (95% CI: 16.2–4.1); *p* = 0.002) and less likely to be divorced/separated or widowed than those in previous surveys (cluster mean difference 10.3% (95% CI: 0.5–20.1; *p* = 0.040)). FSW in 2021–2023 were more likely to have started sex work before the age of 20 (38.9% cluster range: 24.4–56.0 compared to 25.8% cluster range: 14.2–37.8; cluster mean difference 13.1% (95% CI: 19.2–6.9), *p*-value < 0.001). They were also more likely to have been in sex work for >5 years (cluster mean difference: 10.5% (95% CI: 16.7–4.3), *p*-value = 0.002). The median number of reported other sex workers known fell from 10 (cluster range: 5–10) in 2016–2017 to 5 (cluster range: 3–10) in 2021–2023. Similar declines were seen for the number of other sex workers seen in the past month (Table 3). When adjusted for site and all sociodemographic characteristics, the change over time in HIV prevalence was of a similar magnitude to the unadjusted analysis (aRR 0.76; 95% CI: 0.67–0.86, *p*-value < 0.001) (Supplementary Appendix A, Table S9). We concluded that while there was some evidence that the characteristics of women engaged in sex work had changed over time, we had no evidence that these changes could explain the HIV prevalence decline observed.

Lastly, we examined FSW engagement in targeted programme services and self-reported HIV risk behaviour (Table 3). Across all

**Table 2 | Respondent-driven sampling diagnostics by pooled surveys**

| RDS diagnostics | Pooled surveys 2016–2017 (N = 4005) | | Pooled surveys 2021–2023 (N = 4475) | | Difference in means (95% CIs) | p-value |
|---|---|---|---|---|---|---|
| | N | Cluster mean [range] | N | Cluster mean [range] | | |
| Homophily for HIV prevalence (survey round)[a] | 4005 | 1.04 [0.91–1.22] | 4475 | 1.07 [0.92–1.16] | N/A | N/A |
| Difference between ratio of egonet & sample proportion <25[b] | | 3.17 [0.0–6.3] | | 3.43 [1.6–5.3] | 0.26 (−1.4–0.8) | 0.630 |
| Difference between ratio of egonet & sample proportion >35[b] | | 1.00 [0.0–1.3] | | 1.00 [0.7–1.4] | 0.00 (−0.2–0.3) | 0.820 |
| | n/N | % | n/N | % | | |
| % recruiters successfully recruiting 2 others[c] | 2100/2646 | 79.0 [71.3–88.9] | 2276/2796 | 79.9 [66.7–89.3] | 0.9 (−1.2–3.0) | 0.410 |
| % seeds with chains reaching the final wave[d] | 106/118 | 90.7 [50.0–100.0] | 81/114 | 69.2 [33.3–100.0] | −21.5 (−31.5–−11.5) | <0.001 |
| HIV prevalence by wave[e] (%) | | | | | | |
| Wave 0 (seeds) | 52/91 | 54.1 [2.2–100.0] | 43/80 | 54.6 [0–100.0] | 0.5 (−24.0–25.0) | 0.970 |
| Wave 1 | 107/179 | 62.1 [39.5–95.4] | 79/176 | 47.8 [10.4–77.2] | −14.3 (−27.7–−0.9) | 0.038 |
| Wave 2 | 211/357 | 59.1 [36.3–82.4] | 130/346 | 41.4 [6.8–79.6] | −17.7 (−30.8–−4.7) | 0.010 |
| Wave 3 | 315/582 | 52.6 [35.0–78.9] | 215/554 | 40.5 [23.2–66.4] | −12.2 (−22.4–−1.9) | 0.020 |
| Wave 4 | 483/879 | 55.1 [28.3–76.3] | 329/786 | 42.6 [27.4 – 69.1] | −12.5 (−22.5–−2.8) | 0.014 |
| Wave 5 | 499/979 | 53.2 [20.6–74.0] | 338/961 | 33.4 [17.0 – 48.4] | −19.8 (−28.5–−11.1) | <0.001 |
| Wave 6 | 277/562 | 44.3 [41.7–46.9] | 356/1065 | 36.1 [3.4 – 100.0] | −8.2 (−22.5–6.0) | 0.240 |
| Wave 7 | 182/376 | 49.0[e] | 144/507 | 37.7 [0–96.3] | – | – |

Source data are provided as a Source Data file. Data are number of participants. *P*-values (two-sided) are based on *t* test.

*RDS* respondent-driven sampling.

[a]Calculated using all FSW included in the samples i.e. without restricting to those aged 18–39 years.

[b]Egonet data was not collected in the 2017 survey.

[c]Calculated using all recruiters that were included in the samples i.e. without restricting to those aged 18–39 years.

[d]*N* is total seeds in the sample i.e. without restricting to those aged 18–39 years.

[e]Only 1 cluster in wave 7 in 2017 – could not calculate range.

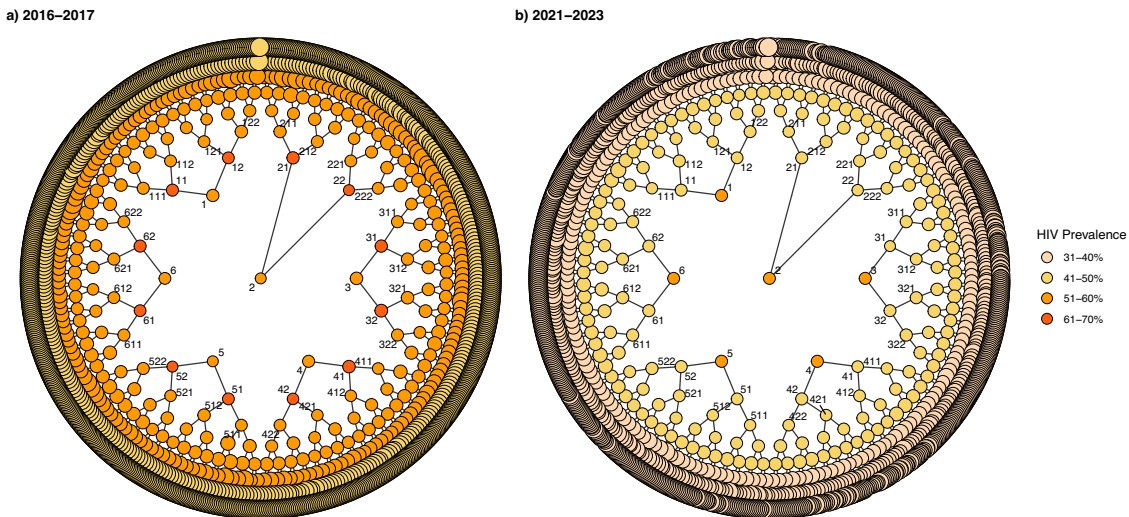

a) 2016–2017     b) 2021–2023

HIV Prevalence
○ 31–40%
○ 41–50%
○ 51–60%
● 61–70%

**Fig. 2 | Combined RDS recruitment trees for 15 pooled survey sites. a** For the period 2016–2017. **b** For the period 2021–2023. Circle 1, for example, depicts the 15 participants who were seed 1 across the 15 sites surveyed in 2016–2017. Circle 2 depicts a total of 15 participants who were seed 2 across the 15 sites, and so on up to circle 6, which depicts a total of 15 participants who were seeds 6 across the 15 sites. The circles connected to these circles 1, 2, 3, 4, 5 and 6, depict participants who were recruited by seeds 1 across the 15 sites and this goes on and on until the final wave 7. The colour coding for HIV prevalence indicates the prevalence in each wave, with everyone in each wave being assigned the same colour. Source data are provided as a Source Data file.

surveys we saw similar HIV testing patterns with a cluster mean in both time periods of >97% FSW reporting having ever HIV tested. By the 2021–23 surveys, some 22.9% of FSW reported currently taking oral HIV pre-exposure prophylaxis, following its introduction in 2017. Reported attendance at outreach meetings was however 12.9% (95% CI: −22.6–−3.1; *p* = 0.012) higher in 2021–2023. Adjusting for programme engagement characteristics (except oral pre-exposure prophylaxis use which was only measured among HIV negative women in later surveys) slightly lowered the prevalence ratio compared to unadjusted models (aRR 0.81; 95% CI: 0.70–0.94, *p*-value < 0.001) (Supplementary Appendix A, Table S9). We concluded that programme engagement remained high across the study period and this, along with the

**Table 3 | Sociodemographic, sex work, risk behaviour and programme engagement characteristics of female sex workers enrolled in 2016–2017 and 2021–2023 respondent-driven sampling surveys**

| Characteristics | Pooled surveys 2016–2017 (N = 4005) | | Pooled surveys 2021–2023 (N = 4475) | | % Difference in means (95% CIs) | p-value |
|---|---|---|---|---|---|---|
| | *n/N* | Cluster mean [range] | *n/N* | Cluster mean [range] | | |
| Socio-demographics characteristics | | | | | | |
| Median age (years) | 4005 | 29.4 years [27–31] | | 28.7 years [25–31] | −0.7 years (−1.8–0.4) | 0.200 |
| Education (%) | | | | | | |
| Primary only or less | 1046/4005 | 30.5 [14.7–49.3] | 831/4475 | 20.3 [13.9–39.4] | −10.2 (−16.3–−4.2) | 0.002 |
| Secondary or higher | 2958/4005 | 69.5 [50.7–85.3] | 3640/4475 | 79.7 [60.6–86.1] | 10.2 (4.1–16.2) | 0.002 |
| Marital status (%) | | | | | | |
| Never married | 854/4005 | 19.4 [9.0–42.1] | 1231/4475 | 25.2 [9.7–55.1] | 5.8 (−3.4–14.9) | 0.210 |
| Divorced/separated/widowed | 3087/4005 | 79.1 [55.1–91.0] | 3045/4475 | 68.8 [39.9 – 86.1] | −10.3 (−20.1–−0.5) | 0.040 |
| Married | 64/4005 | 2.0 [0.1–5.9] | 199/4475 | 6.1 [0.6 – 14.0] | 4.1 (1.9–6.4) | 0.001 |
| Sex work characteristics | | | | | | |
| Started selling sex at an age <20 years (%) | 1071/4005 | 25.8 [14.2–37.8] | 1856/4475 | 38.9 [24.4–56.0] | 13.1 (6.9–19.2) | <0.001 |
| >5 years since starting sex work (%) | 1548/4005 | 39.9 [31.3–58.4] | 2275/4475 | 50.4 [34.5–65.2] | 10.5 (4.3–16.7) | 0.002 |
| Median no. of other sex workers known | 4005 | 12 [7–20] | 4475 | 5.5 [4–8] | −6.5 (−8.8–−4.2) | <0.001 |
| Median no. of other sex workers seen in the past month | 4005 | 10.1 [6–15] | 4475 | 4.1 [3–5] | −5.9 (−7.4–−4.4) | <0.001 |
| Median no. of clients in the past week[a] | 4005 | 5.8 [4–10] | 4475 | 4.9 [3–6] | −0.9 (−1.9–0.1) | 0.080 |
| Median no. of steady partners in the past year[b] | 4005 | 1.0 [1.0–1.0] | 4475 | 3.5 [1–6] | 2.5 (1.8–3.2) | <0.001 |
| Risk behaviour characteristics | | | | | | |
| Condomless sex[c] (%) | 2014/4005 | 54.8 [44.0–70.8] | 3204/4475 | 73.8 [62.3–87.2] | 19.0 (13.4–24.6) | <0.001 |
| STI symptoms[d] (%) | 1345/4001 | 35.4 [18.6–58.9] | 2482/4475 | 58.3 [32.3–71.0] | 22.9 (14.9–30.9) | <0.001 |
| STI biomarker (%) | | | | | | |
| *Gonorrhoea (NG)* | 53/1214 | 3.6 [2.2–5.1] | 576/3125 | 21.0 [5.2–45.2] | 17.4 (10.2–24.6) | <0.001 |
| *Chlamydia (CT)* | 54/1214 | 4.3 [4.0–4.7] | 724/3125 | 23.8 [16.9–43.8] | 19.5 (15.1–23.8) | <0.001 |
| *Trichomoniasis (TV)* | 150/1221 | 12.2 [11.6–12.8] | 1003/3125 | 34.3 [12.4–76.3] | 22.1 (13.1–31.2) | <0.001 |
| PrEP[e] (%) | n/a | n/a | 755/2841 | 22.9 [11.7–37.3] | – | – |
| Service engagement characteristics | | | | | | |
| Ever HIV tested (%) | 3942/4005 | 97.9 [95.4–100.0] | 4343/4475 | 97.4 [93.5–99.8] | −0.6 (−1.9–0.8) | 0.390 |
| Last HIV test within 3 months (%) | 1774/3942 | 45.9 [30.3–63.6] | 2034/4343 | 47.5 [34.1–61.1] | 1.6 (−4.1–7.4) | 0.560 |
| Attended CeSHHAR clinic in the past 12 months[f] (%) | 2004/4005 | 61.1 [23.2–82.0] | 2172/4475 | 54.1 [26.2–69.1] | −7.1 (−18.2–4.1) | 0.200 |
| Attended outreach meeting in the past 12 months[g] (%) | 1290/4005 | 37.6 [14.9–63.4] | 2109/4475 | 50.4 [21.6–65.4] | 12.9 (3.1–22.6) | 0.012 |

Source data are provided as a Source Data file. Data are number of participants. *P*-values (two-sided) are based on *t* test.
[a]In the past week, how many clients have you had sex with?
[b]How many steady partners have you had in last 12 months (include current partner if you have one)?
[c]Condomless sex - in the past month, how often did you use condoms with your commercial partner/client and with your permanent/steady partner.
[d]Self-report of at least one STI symptom. Symptom options: bad-smelling, abnormal genital discharge, genital sore or ulcer, pelvic pain, genital sores or itching, genital warts or an unusual vaginal discharge, any disease got through sexual contact.
[e]PrEP provision started in the programme in 2017 - data were not collected on PrEP until 2021.
[f]In the past 12 months, have you attended the Sisters with a Voice/CeSHHAR clinic. In 2017 this was asked for the past 6 months i.e. in the past 6 months, have you attended the Sisters with a Voice clinic.
[g]In the past 12 months, have you been to one or more meetings where a sex-worker peer educator/empowerment worker has spoken (meeting means with more than one sex worker)? In 2017 this was asked for the past 6 months i.e. in the past 6 months, have you been to one or more meetings where a sex-worker peer educator/empowerment worker has spoken (meeting means with more than one sex worker)?

introduction of oral pre-exposure prophylaxis, may have made a partial contribution to the measured HIV prevalence decline.

The reported number of clients in the past week was marginally higher in 2016–2017, but the reported number of regular partners in the past year was three times higher in 2021–2023 (2016–2017 cluster mean: 1.0 vs 2021–2023 cluster mean 3.5; *p*-value < 0.001). Condomless sex, measured as not having used a condom with clients and steady partners in the previous month, was 19% (95% CI: 24.6–13.4; *p* < 0.001) higher in 2021–2023 with a cluster mean of 73.8% compared to 2016–2017 where this was 54.8%. Self-reported STI symptoms increased between survey rounds from 35.4% to 58.3% (cluster mean difference 22.9% (95% CI: 30.9–14.9, *p* < 0.001). STI prevalence measured with biomarkers for gonorrhoea (NG), chlamydia (CT) and Trichomoniasis (TV) all increased by over 17% (*p* < 0.001) between 2017 and 2023. As with our other models there remained a significant reduction in prevalence ratio of HIV when we adjusted for condom use and STI symptoms (aRR 0.69; 95% CI: 0.61–0.78) (Supplementary Appendix A, Table S9). We concluded that individual HIV risk behaviour did not appear to have improved between 2016–2017 and 2021–2023, and we therefore had little evidence that changes in risk behaviour could explain the HIV prevalence decline that we observed.

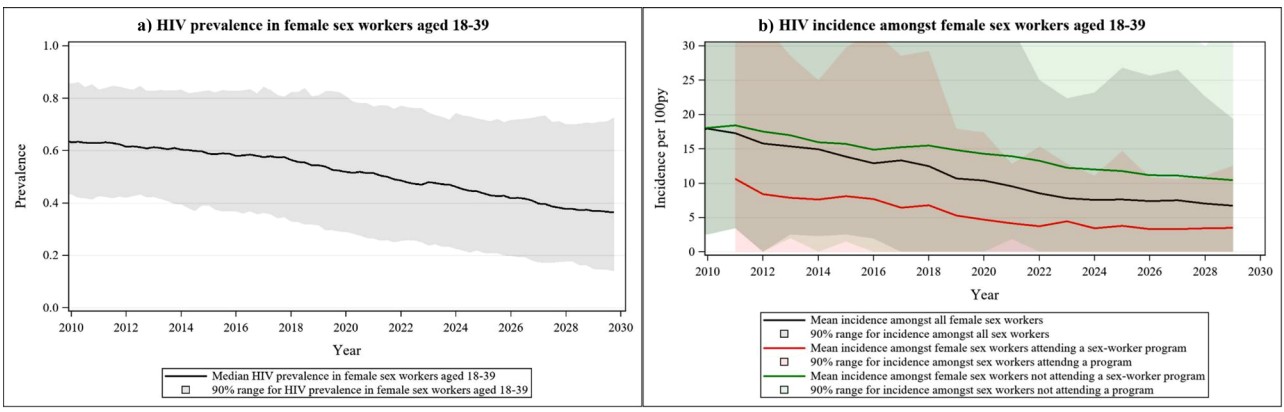

**Fig. 3 | Modelled HIV prevalence and incidence estimates. a** Modelled HIV prevalence from 2010 to 2030 among female sex workers aged 18–39 years. **b** Modelled HIV incidence from 2010 to 2030 among female sex workers aged 18–39 years.

### Triangulation with modelled HIV prevalence & incidence

We modelled HIV prevalence and HIV incidence among FSW between 18–39 years using the HIV Synthesis model calibrated to Zimbabwe[29]. Modelled HIV prevalence among 18–39 year old FSW showed a steady decline over time, of a smaller magnitude to that observed in the empirical data, falling from 58.0% (90% range: 36.8–82.9%) in 2016 to 51.5% (90% range: 26.5–76.9%) in 2021 and 48.0% (90% range: 23.8–74.6%) in 2023 (Table 1, Fig. 3 & Supplementary Appendix A, Supplementary Figs. S11–13). Modelled HIV prevalence among FSW was higher than estimates from the RDS survey data at the majority of the time points (Supplementary Appendix A, Supplementary Fig. S11). Modelled HIV incidence fell from 12.9 (1.9–30.8) per 100py in 2016 to 9.5 (0.0–32.8) per 100py in 2021 among 18–39 year olds (Fig. 3a). HIV incidence was lower when modelled in the context of a FSW programme, from 2010 onwards, declining from 7.7 (0.0–32.2) per 100py in 2016 to 4.1 (0.0–12.8) per 100py in 2021 (Fig. 3b).

## Discussion

HIV prevalence among FSW in Zimbabwe declined significantly, by 15.7%, between 2016–2017 and 2021–2023, having previously been stable over several years[22,30]. HIV prevalence increased steeply with age in all surveys, consistent with high HIV incidence. However, declines in HIV prevalence among most age cohorts between 2016 and 2021 suggested lower HIV incidence in later time periods. Modelled estimates also suggested a decline in HIV prevalence and incidence among FSW over time. There was no evidence that RDS-recruitment biases or changes in the sociodemographic characteristics of women selling sex could explain the HIV prevalence decline observed. Individual-level behavioural risk appeared, if anything, to increase rather than decrease over the study period. As discussed further below, HIV treatment expansion among men living with HIV, coupled with high programme engagement and the introduction of pre-exposure prophylaxis (PrEP) among FSW, are likely drivers of the lower HIV incidence among FSW in recent years.

Our research platform is unique in Southern Africa and our study provides novel, robust information on trends in HIV prevalence among FSW in Zimbabwe. We report a sharp population-based decline in HIV prevalence, with empirical data and mathematical modelling consistent with recent HIV incidence declines over the same period[5]. Our estimates of changes over time are unlikely to suffer from Hawthorne effects noted in previous cohort studies of FSW[29]. However, it is possible that some of the responses we received in the earlier years, when we used interviewer-administered personal interviewing suffered from reporting bias[31–33]. Risk behaviours may have been underreported, and programme engagement overreported due to social desirability[31–33]. Further, despite our use of a robust approach to collecting behavioural

risk data with audio computer-assisted self-interviewing in the later years, which enhances privacy and promotes participants' comfort and honesty, it is likely that some reporting bias in relation to explanatory factors remained[34,35].

We have used a standardised and robust approach to surveying FSW over more than 10 years. Across multiple locations and time periods, RDS diagnostics suggested the sampling approach worked consistently. In both 2016–2017 and 2021–2023 surveys, over 75% of recruits successfully recruited two other participants, although a lower proportion of recruitment chains reached the final wave in 2021–2023. We cannot rule out that in later surveys recruitment chains may have included participants from a wider geographic area due to the broadening geographic scope of the KP programme, and it is possible this may have led to recruitment from a lower risk pool of FSW. While RDS has limitations, our application of a standardised methodology accompanied by the RDS diagnostics we present, suggest that sampling bias cannot explain the HIV prevalence decline we observed.

A limitation of our analysis is that we were unable to directly measure HIV incidence among FSW and can only indirectly assess whether declines in the rate of new infections among FSW explain the changes in prevalent infections we observed. In line with previous work, we consistently observe a strong increase in HIV prevalence by age which is consistent with high HIV incidence[30]. We also explored HIV prevalence changes over time among age cohorts, following an approach developed by Hallett et al. This "synthetic cohort" approach has been used to estimate HIV incidence from serial cross sectional surveys, making assumptions about likely levels of mortality[26–28]. However, in the case of FSW, the effect of HIV-mortality is likely to be dwarfed by the potential rates of women starting and stopping sex work. Little is known about HIV prevalence at entry and exit from sex work and so we could not calculate HIV incidence from cohort-specific changes in HIV prevalence. Further, entry and exit patterns from sex work may have changed during the study period, which coincided with the COVID pandemic and major economic disruption. Nevertheless, relative changes in prevalence between years within age cohorts are likely to be correlated with HIV incidence, and these patterns suggested lower HIV incidence over time. Further, small increases or decreases in HIV prevalence within age cohorts, as we observe here, are unlikely to be compatible with very high levels of HIV incidence among FSW.

In the absence of direct measures of HIV incidence, we triangulated our observations with the results of a mathematical model calibrated to Zimbabwe. HIV prevalence was lower in our empirical studies than predicted by the HIV synthesis model, though trends over time were similar. Since self-reported data on sexual behaviour and condom use are subject to social desirability bias, the model reflects

uncertainty in these parameters by sampling key parameters, resulting in wide uncertainty ranges. The modelled estimates of HIV prevalence are largely in line with empirical estimates on FSW seen under the KP programme in Zimbabwe, and it is not clear whether modelling assumptions or biased estimates are more likely to explain any differences observed between the model and the observed data. Even with a modelled decline in prevalence lower than observed from the surveys, the model estimates a substantial decline in underlying incidence.

We have previously used a variety of methods, each with their own limitations, to directly estimate HIV incidence among Zimbabwean FSW[5,15,30]. Evidence is accumulating that HIV incidence among FSW is now declining. In a previous analysis of HIV testing data collected through service delivery within the KP programme, we documented a declining rate of seroconversion among programme attenders[5]. Results from LAg avidity and viral load testing in a recent infection testing algorithm (RITA) used to estimate HIV incidence among programme attenders and participants in our 2021 RDS survey suggested lower HIV incidence than expected on the basis of previous work[21], although with wide confidence intervals[36].

A 2023 review found that relative HIV incidence among women who engage in sex work in sub-Saharan Africa remained constant over time compared to modelled incidence among women in the wider population, indicating overall declines in HIV incidence among women who engage in sex work in the region. In the same review, case studies in Kenya and Zimbabwe showed different country-level trajectories. In Kenya, relative declines in HIV prevalence were similar among women who engage in sex work and the wider female population, while in Zimbabwe the relative declines among FSW were slower compared to the steeper declines among women in the wider female population[3]. The case studies did not include recent data as we did here, which likely explains the apparent slower decline they found among women who engage in sex work in Zimbabwe.

We explored possible underlying drivers of recent changes in the epidemiology of HIV among FSW in Zimbabwe. One possible explanation might be changes in the population of women engaged in sex work. On average, women in sub-Saharan Africa spend 3–4 years in sex work[37], and our study tracked trends between surveys with gaps of 3–6 years between them. Women recruited in 2021–2023 were more likely to have completed secondary education and less likely to be divorced/separated or widowed than women recruited in 2016–2017, both of which may be associated with lower HIV prevalence. However, there is limited quantitative evidence available on the extent to which HIV status or other factors are associated either with the propensity to start sex work, or to leave it[38,39]. Additionally, we were not able to assess the role of other drivers of HIV prevalence such as mortality, transitions into and out of sex work, and migration. Mortality might not have had a significant impact, as viral suppression has improved significantly over time among FSW, likely positively impacting survival rates. More research is needed to understand the potential impacts of population turnover resulting from these drivers on estimates of HIV incidence among FSW. Our data align with modelling from South Africa, showing that sex work dynamics are likely to be changing and the duration at risk is getting longer[40,41]. By 2021–2023 a greater proportion of FSW had started in sex work before the age of 20 and had been in sex work for more than 5 years. However, women in our later surveys were of similar age to those in earlier years.

In addition, the period over which we have shown major declines in HIV prevalence coincided with the emergence and global spread of SARS-CoV-2, and associated lockdowns and other containment measures. Several studies have reported that the COVID-19 pandemic had a profound but complex effect on sex work[42,43]. Lockdowns may have reduced numbers of clients, but may have also compounded the economic conditions that drive women into sex work[44]. Interestingly, women in our study reported knowing fewer other sex workers in later survey rounds. This may indicate changes in sex worker networks, possibly due to shifts in patterns of entry and exit from sex work or a transition to phone or internet-based sex work post-COVID pandemic. Using data from two of the studies included here, we have estimated that the proportion of adult women engaged in sex work in Zimbabwe increased from 1.2% to 1.6% between 2017 and 2023 with overlapping plausibility bounds. The entrance of more women into sex work may have diffused HIV risk and lowered HIV prevalence, as less networked communities may have fewer shared clients or regular sexual partners, impacting HIV transmission dynamics[45]. Drawing on the results of adjusted analysis, we ultimately judged that changes in the sex work population cannot explain all of the decline in HIV prevalence we have observed here, but future monitoring of these dynamics is essential.

Consistently high programme engagement, particularly the increase in the proportion of FSW reporting attendance at outreach meetings, may have contributed to the decrease in HIV prevalence in the later years. Having a programme in place[3], whether at a lower or higher intensity level, has been found associated with lower HIV incidence compared to discontinuing the programme[46]. Changes in the sexual risk behaviour or use of prevention interventions of FSW might also underpin lower HIV incidence in later years. Oral pre-exposure prophylaxis became available in 2017, and by the 2021–2023 surveys some (22.9%) of the HIV negative FSW reported currently being on PrEP. While this offers a potential explanation for lower HIV incidence, we were not able to assess this in adjusted analysis as PrEP data were not available in early years and are only asked about among HIV-negative FSW. In our recent (Adapted Microplanning to Eliminate Transmissible HIV in Sex Transactions) AMETHIST trial, despite high levels of self-reported PrEP use we found that <1% of those who self-reported PrEP use had protective levels of tenofovir[24]. The self-reported number of clients in the past week was lower, but the number of steady partners was higher, suggesting that FSW were likely to have multiple partners with whom they could have condomless sex. This may explain why condomless sex was more commonly reported in later surveys and is consistent with other studies conducted in sub-Saharan Africa where FSW were more likely to have condomless sex with steady partners than clients[47,48]. While self-report may be biased, it is unlikely that this bias differed between surveys and the difference was substantial at ~20%. Self-reported STI symptoms and STI prevalence were also higher which would suggest more, rather than lower, risk behaviour in later surveys[49,50]. The overall pattern we observed appeared unlikely to be consistent with significantly lower behavioural risk as a driver of lower HIV prevalence or incidence, although oral PrEP expansion may have played a role.

Another limitation of our study is the absence of data on HIV prevalence and antiretroviral treatment among the male sexual partners of FSW. One plausible explanation for the decline in HIV prevalence we observed here may be an increase in the coverage of antiretroviral treatment, and therefore viral suppression, among male partners of FSW, in turn leading to lower HIV transmission probabilities. Additionally, the significant improvements of viral suppression among FSW over time, may have potentially reduced the risk of transmission to HIV-negative male partners, which in turn may have further decreased the likelihood of HIV-negative FSW becoming infected by male partners. While ART programmes are known to have been more successful in women than men, the PHIA surveys in Zimbabwe have shown an increase in suppressed HIV infection from 48.9% to 68.1% among adult (15–49 year old) men between 2016 and 2020, driven by treatment expansion[2,51]. This reflects an ongoing need for strengthening treatment among men, but also reflects an increase in treatment access in recent years. In common with other populations and settings, we judge that increased testing and antiretroviral treatment uptake may have been a driver of HIV incidence decline among FSW[52]. This study assessed this important possibility among FSW and further research in this area is essential.

In conclusion, we present data from Southern Africa that population level HIV prevalence has declined among FSW. We suggest that this is likely attributable in part to lower rates of new infection driven by increased antiretroviral treatment uptake among FSW themselves and men, and consistently high levels of programme engagement coupled with the availability of oral pre-exposure prophylaxis. However, there is limited evidence that reduced risk behaviour among FSW has yet been a primary driver of HIV incidence declines and there is room for cautious optimism about these trends, but also a critical need to continue to monitor and strengthen HIV prevention among FSW. We are concerned that there remains a fatalism among FSW with respect to their capacity to avoid contracting HIV[53]. In addition, the important but intense focus on treatment literacy and expansion in recent years may have sidelined efforts to fully promote HIV prevention. In the coming years, we will re-focus our efforts on primary HIV prevention messaging, expand the toolbox of prevention options for FSW and continue to find ways to foster community leadership of the programme. Sustaining HIV treatment access for men living with HIV is also essential. If achieved, our data suggest these actions will further drive down HIV risk among women selling sex in Zimbabwe and the wider region.

## Methods

### Study setting

The Centre for Sexual Health and HIV/AIDS Research (CeSHHAR) Key Populations Programme (formerly Sisters with a Voice) was established in Zimbabwe in 2009 on behalf of the Ministry of Health and Child Care (MoHCC) and the National AIDS Council (NAC). By 2023, the programme operated nationally, in 40 districts across all 10 provinces in Zimbabwe. Services are delivered through a primary care clinic serving a site that is a geographical catchment area identified as a known hotspot for sex work. Sites are urban, rural and highway locations and either have a static clinic (14 sites in 2024) and drop-in centres (14 in 2024) delivering services 5 days a week, or a mobile outreach clinic (126 in 2024) delivering services at least one day a week. At each site, comprehensive sexual and reproductive health services are delivered in line with WHO guidelines[54–56]. These services include free condoms and contraception, provider-initiated HIV testing and counselling, HIV self-testing and counselling (and secondary distribution of self-test kits for partners), syndromic management of sexually transmitted infections (STIs), health education and legal advice supported by a network of peer educators. Additionally, clinics provide long-acting reversible contraception (implants), referral for cervical cancer screening and on-site access to pre-exposure prophylaxis and, since 2020, antiretroviral therapy and viral load testing.

### Respondent-driven sampling surveys

Between 2013 and 2023, we conducted five multi-site RDS surveys in locations served by the programme. Surveys in 2013 and 2016 were conducted in 14 programme sites as part of a programmatic impact evaluation, known as the Sisters Antiretroviral Programme for Prevention of HIV, an Integrated Response (SAPPH-IRe) trial[23]. In 2017, surveys were conducted in three sites including Zimbabwe's two main cities; Harare and Bulawayo, as part of an exercise to generate a national population size estimate (PSE) for FSW[57]. In 2021, surveys were conducted in 22 sites participating in the AMETHIST cluster randomised trial[20,24]. In 2023 we conducted second PSE surveys in Harare and Bulawayo, two of the same cities as the 2017 PSE survey. Surveys conducted in 2013, 2016 and 2021 were in the same 13 programme sites, while surveys conducted in 2017 and 2023 were in the same two programme sites (both major cities) (Supplementary Appendix A, Table S1).

From 2013 to 2023, each survey round maintained similar procedures, including social mapping followed by the purposive selection of initial participants (seeds) representing diverse ages, types of sex work, and geographic locations (Supplementary Appendix A, Table S1). The social mapping and selection of seeds were largely similar across surveys in 2013, 2016, 2017 and 2023, involving visits to sex work venues (e.g. bars, touchlines) within the urban districts of a programme site and conversations with individual FSWs. In 2021, the geographical reach of some programme sites increased and seed selection was expanded to include the peri-urban districts. Seeds were recruited in 2021 through focus group discussions coordinated by programme micro-planners. We interviewed these seeds, collected a blood sample for HIV testing, and provided them with two coupons to distribute among their peers. Women who received a coupon were invited for an interview and then given two coupons to pass on to their peers. We repeated this process for five to seven iterations (waves) exclusive of the seeds. Participants received a US$5 compensation for their time and an additional US$2 for each eligible and recruited referral. Upon arrival for the survey, an informal verification to ensure participants were sex workers and had not previously participated was carried out by Research Assistants. We used a coupon management system to ensure that coupons were genuine and to minimise repeat participation.

Data collection methods evolved over time. From 2013 to 2017 we collected interviewer-administered questionnaire data on tablet computers, which included demographics, sex work, sexual behaviour, HIV prevention, care uptake, and personal network size for RDS adjustment. In 2021 and 2023, we collected these data using audio-computer assisted self-interviewing. HIV testing procedures were the same across surveys as adapted from Zimbabwe's national testing and counselling guidelines. Determine HIV-1/2 or First Response HIV-1-2 kit antibody testing was used as the first screening test. Where the result was HIV positive, this was confirmed using First Response HIV-1-2 kit or Determine HIV-1/2.

### Statistical analysis

**Site and participant inclusion.** For our current analysis, we included the same study sites surveyed in multiple survey rounds. This included 13 programme sites (towns) in 2013, 2016 and 2021 and another two programme sites (cities) surveyed in 2017 and 2023. Sites were included only if they were surveyed in either 2013, 2016 and 2021 (13 towns), or 2017 and 2023 (2 cities). Individuals were included if they were aged 18–39 years and were not missing data on RDS recruiter information and on HIV testing. We focused on women under 40 years of age since HIV prevalence changes are more likely to be reflective of HIV incidence patterns, and because of data sparsity at older ages. All other sites surveyed in these years were excluded from this analysis. As shown in Supplementary Appendix A, Table S1, the analysis in this paper was conducted on a subset of sites and of individuals who were recruited to these multi-site surveys. We appended data from all survey rounds for our analytical dataset (Supplementary Appendix A, Table S1).

**Analysis of multi-site RDS surveys.** Our approach to RDS analyses for individual surveys have been described in detail elsewhere[16,23,24,57]. When reporting the specific studies, we followed the STROBE-RDS guidelines[22,24,58], first describing the population samples recruited at each site and assessing the evidence of bias in our operationalisation of RDS[59]. We used the RDS-II estimator[60] for analysis: dropping seed responses and weighting each woman in each site by the inverse of her network size, that is, the number of other women she reports knowing and could have recruited. For this analysis, observations were weighted using site-normalised inverse degree weights, accounting for the original recruitment structure in each study site.

**HIV prevalence over time and by age.** Firstly, we calculated crude and RDS weighted HIV prevalence for each survey (2013, 2016, 2017, 2021 and 2023) and for the main analysis, we pooled estimates for

2016–2017 and 2021–2023. To account for the clustered nature of our data by site, we calculate prevalence estimates for each site (cluster), sum them, and divide by the number of sites ($n = 13$ for 2016–2017 pooled data and $n = 2$ for 2021–2023 pooled data). We also calculated the minimum and maximum estimates (range) across 13 sites for 2016–2017 pooled data and 2 sites for 2021–2023 pooled data. We compared the difference between our pooled estimate site (cluster) mean for 2016–2017 data and 2021–2023 data by using a *t*-test and obtained 95% confidence intervals (CI) and a *p*-value.

Secondly, to explore the potential underlying trends in HIV incidence we examined age-specific HIV prevalence, both within and between individual survey rounds. For each survey we calculated cluster mean RDS-weighted age-specific HIV prevalence and plotted these for the same 13 sites for years 2013 and 2016, and for 2016 and 2021, as well as for 2 different sites for 2017 and 2023. We also present age-specific HIV prevalence for pooled survey rounds which each included the same 13 + 2 sites (2016–2017 and 2021–2023) and calculated 95% CI and *p*-value for the difference in cluster means. Our approach was based on age-specific prevalence analysis that has previously been reported for pooled data from RDS surveys conducted between 2011 and 2016[30].

Thirdly, we used an approach adapted from Hallett et al.[26–28] to present age-specific HIV prevalence estimates over multiple survey rounds. We calculated HIV prevalence for each year of age by survey round and present the age cohort cluster mean and range across surveys. We present the group a FSW was in during an earlier survey round versus her corresponding age group in the following survey round if she had been surveyed again. We present this data for the same 13 sites (towns) surveyed in 2013, 2016, and 2021, and then separately for the two sites (cities) surveyed in 2017 and 2023. We present data for each survey round separately to account for the varying lengths of time between the surveys, which would not have been possible to capture if our 2016–2017 and 2021–2023 data were pooled. For example, a sex worker who was 18 years old in the 2016 survey would be 23 years old in the 2021 survey.

**RDS diagnostics.** We hypothesised that a potential explanation for the decline in estimated HIV prevalence between the survey rounds might be that the network sampling process inherent to RDS operated differently across surveys. For each survey in each location at each time point we ran a set of recognised RDS diagnostic tests intended to provide some information on whether the random walk process that theoretically underpins RDS worked appropriately and if seed dependence was sufficiently removed[59]. Firstly, we generated the recruitment trees by HIV status for each site by year, and for pooled 2016–2017 and 2021–2023 surveys by HIV prevalence, and calculated the percentage of recruiters successfully recruiting two other FSW to the survey excluding the final wave, to understand both the forward recruitment process and the composition of the sample (Fig. 2 & Supplementary Appendix A, Supplementary Fig. S1–S10). Secondly, we generated the combined convergence and bottleneck plots of the estimated HIV prevalence over sample accumulation, calculated the percentage of seeds with recruitment chains reaching the final wave, and estimated HIV prevalence by wave of recruitment to assess whether the estimate appeared to have stabilised over successive waves of recruitment and was independent of initial seed characteristics as well as to show the dynamics of the estimates from each individual seed (bottlenecks) (Supplementary Appendix A, Supplementary Figs. S1–S10). Thirdly, we calculated recruitment homophily with respect to HIV prevalence to understand if HIV positive FSW were more likely to recruit HIV positive peers. We then calculated the ratio of the proportion of FSW < 25 years old and >35 years old in the sample, compared to the proportion of individuals <25 years old and >35 years old who were self-reported by participants to be in their social networks (egonet). As with our main prevalence estimates, we subsequently calculated the cluster means for

other RDS diagnostics namely the percentage of recruiters successfully recruiting two other FSW to the survey excluding the final wave, the percentage of seeds with recruitment chains reaching the final wave, HIV prevalence by wave of recruitment and homophily for HIV prevalence, by survey round and for pooled survey rounds (Table 2 & Supplementary Appendix A, Supplementary Fig. S6) so we could use these to compare potential reasons for any changes over time.

**Sociodemographic, sex work, risk behaviour & programme engagement characteristics.** We hypothesised that a potential explanation for the decline in estimated HIV prevalence between surveys might be that over time women with different background sociodemographic characteristics related to their risk of HIV infection may have entered or exited sex work, and thus that the sociodemographic composition of the sample may differ across years. We calculated cluster summary statistics for sociodemographic variables in each survey and for our pooled survey years, as described earlier. For pooled surveys, we calculated the difference between cluster means. We explored sociodemographic characteristics of age, education and marital status. We also hypothesised that differences in sex work characteristics may have played a role in HIV prevalence decline, specifically the age at starting sex work and years since starting sex work, and the number of other sex workers known and seen in the past month. We explored difference in service engagement between survey rounds analysing variables on HIV testing (ever tested for HIV and tested within the last three months) and engagement in CeSHHAR's KP programme: clinic attendance in the past 12 months and attendance at a peer-led outreach meeting in the past 12 months. Lastly, we hypothesised that a potential explanation for the decline in estimated HIV prevalence between surveys might be that women more commonly practiced safe sex behaviours or were effectively using HIV prevention products. We compared the number of clients women reported in the past week, the number of steady partners, condomless sex in the previous month with any partner, self-reported STI symptoms and PrEP use.

**Adjusted analysis.** Lastly, we ran an adjusted analysis using modified Poisson regression to compare our pooled HIV prevalence estimates. In all our models we include programme site as a fixed effect. We then adjusted our model separately for sociodemographic factors, sex work specific factors, risk behaviours and HIV service engagement factors. Our analyses were conducted in R programming language (version 4.4.2).

**Modelled HIV prevalence and HIV incidence**
In addition to our analysis of empirical RDS survey data we modelled estimates of HIV prevalence and HIV incidence for FSW in Zimbabwe over the same time period, including the presence of a sex worker programme in the model from 2010. HIV Synthesis is an individual-based simulation model of HIV transmission, progression and the effect of ART that has been previously described[29]. In brief, each time the model is run, a simulated population of adults is generated (Supplementary Appendix B). Variables are updated every 3 months from 1989 and include age, sex, the number of short term condomless sex partners, presence of a long term condomless sex partner, use of PrEP, HIV testing, HIV acquisition and additionally, in people living with HIV, viral load, CD4 count, use of specific ART drugs, adherence to ART, resistance to specific drugs and risk of HIV-related death (Supplementary Appendix B). All women in the model have a risk of initiating sex work based on several factors including age, overall population levels of sexual risk behaviour and previous experience of being a sex worker. A sex worker programme was modelled from 2010 onwards. The programme was assumed to reduce the likelihood of FSW interrupting treatment and being lost at diagnosis and improve adherence, to reflect for example the impact of ART counselling. Sex worker

programmes were also assumed to provide the following additional benefits: 6-monthly HIV testing, reduction in the number of condomless partners (i.e. greater condom use) and increased willingness to take PrEP. Key model outputs on FSW are shown in Supplementary Appendix A, Table S12, along with empirical estimates from the AMETHIST study. While the model was not directly calibrated to FSW incidence (this is difficult to align with FSW prevalence along with other key epidemic characteristics e.g. 90-90-90s), on the whole, modelled estimates were in line with those reported in the AMETHIST study.

### Ethical approvals

The SAPPH-IRe trial surveys conducted in 2013 and 2016 received ethical approval from the Medical Research Council of Zimbabwe (Ref: MRCZ/A/1762), the London School of Hygiene and Tropical Medicine (Ref: 6524) and the University College London (Ref: 4948/001). The PSE 2017 surveys were approved by the Medical Research Council of Zimbabwe (Ref: MRCZ/A/1942), the London School of Hygiene and Tropical Medicine (Ref: 9686), the University College London (Ref: 6084/003) and RTI International (Ref: 13975). The AMETHIST trial surveys conducted in 2021 were approved by the Medical Research Council of Zimbabwe (Ref: MRCZ/A/2559), the London School of Hygiene and Tropical Medicine (Ref: 19123), and the Liverpool School of Tropical Medicine (Ref: 19-115). The PSE 2023 surveys were approved by the Medical Research Council of Zimbabwe (Ref: MRCZ/A/2867) and the London School of Hygiene and Tropical Medicine (Ref: 26849). Written informed consent, in English, Shona or Ndebele, was obtained for survey participation before interviews were done and blood samples were taken.

### Reporting summary

Further information on research design is available in the Nature Portfolio Reporting Summary linked to this article.

## Data availability

The analytical data generated by pooling relevant variables from 2013, 2016, 2017, and 2023 RDS surveys for this study have been deposited in the Zenodo database and are publicly available at https://zenodo.org/records/15857385. The original RDS datasets are protected to ensure participant privacy and are not publicly available. However, researchers outside of CeSHHAR who are conducting studies with clearly defined objectives may request access to original data, including de-identified individual participant data and the data dictionary used in the studies. Requests should be submitted to the CeSHHAR Data Protection Officer at dataprotection@ceshhar.org cc'ing the first author at sungai.chabata@ceshhar.org. Subject to the submission of a written proposal and a signed data sharing agreement, prespecified data will be made available within 15 working days. Source data are provided in this paper. Source data are provided with this paper.

## Code availability

We have made the data processing and analysis R script publicly available in an online repository (https://zenodo.org/records/16948766).

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

## Acknowledgements

We would like to thank all the female sex workers who participated in respondent-driven sampling surveys and CeSHHAR research staff for their roles in the operationalisation of the survey procedures over the years. We thank the Ministry of Health and Child Care, and the National AIDS Council for supporting our research activities across the country. S.T.C., T.H. and F.M.C. were partly or wholly funded by the Gates Foundation (INV-061640) and (INV- 066092). S.T.C., H.S.J., T.H., M.S.A., L.B-M., F.M., R.S., J.B., A.N.P., F.M.C. and J.R.H. were partly or wholly funded by the Wellcome Trust (214280/Z/18/Z). F.M.C. was partly or wholly funded by the National Institute of Maternal Health (R34 MH129220-01). F.M.C. and J.R.H. were partly or wholly funded by the Gates Foundation (INV-007055). A.T., P.M. and J.D. were partly or wholly funded by the President's Emergency Plan for AIDS Relief through the United States Agency for International Development (award no. 72061320CA00008). M.S.A. was partly or wholly funded by the US National Institutes of Health (5R01MH114560–03), the Gates Foundation (INV-033650), US National Institutes of Health Research (NIHR 301634), and the Wellcome Trust (082384/Z/07/Z). R.Y. was partly or wholly funded by the National AIDS Council. O.M. was partly or wholly funded by the Ministry of Health and Child Care. A.N.P. and L.B-M. were partly or wholly funded by the Gates Foundation (INV-007145) and EU (WT: 6727742).

## Author contributions

S.T.C. and J.R.H. conceptualised and designed the study. F.M.C., J.R.H. and J.B. contributed to funding acquisition. S.T.C., F.M., A.T., P.M. and J.D. contributed to data collection. H.S.J., T.H. and M.S.A. conducted respondent-driven sampling data analysis. S.T.C., M.S.A. and T.S. conducted respondent-driven sampling diagnostic analysis. L. B-M. and A.N.P. contributed to model development and coding and ran the models. S.T.C., H.S.J., T.H., M.S.A., L.B-M., A.T., P.M., J.D., R.S., J.B., R.Y., O.M., A.N.P., F.M.C. and J.R.H. interpreted the data. H.S.J., T.H., M.S.A., A.T. and J.D. contributed to data management. S.T.C., H.S.J., L. B-M. and J.R.H. wrote the original manuscript draft. All authors reviewed the manuscript.

## Competing interests

The authors declare no competing interests.
