## [Transparent Peer Review file · Nature Communications]

Decline in HIV prevalence among female sex workers in Zimbabwe between 2013 and 2023

Corresponding Author: Professor Frances Cowan

Version 0:

Reviewer comments:

Reviewer #1

(Remarks to the Author)

The authors of this study aimed to examine trends in HIV prevalence among female sex workers (FSW) in Zimbabwe over time and to determine whether the anticipated decline in prevalence reflects a true decline in incidence. They analyzed data from five RDS surveys conducted between 2013 and 2023. To assess whether the observed decline in prevalence was due to declining incidence, the authors performed several exploratory analyses, including:

- Comparing HIV prevalence by birth cohort across surveys.
- Investigating potential biases in RDS recruitment over time.
- Examining changes in FSW demographics, risk behaviors, and engagement with HIV care.

The authors conclude that these analyses—collectively— suggest a decline in both HIV prevalence and incidence over time, a finding further supported by a parallel modeling analysis conducted by the authors.

I found the manuscript difficult to follow due to the lack of a clear central analysis and the presence of too many small, descriptive analyses. The methodology was not always well-explained—for instance, the trend analysis of HIV prevalence across different years and locations was difficult to interpret. Additionally, the simulation model used as comparison lacked sufficient methodological details to understand and evaluate and appeared out of place. Given the complexity of a model, a dedicated stand-alone paper with more details on model development and calibration would have been more appropriate. Lastly, since changes in prevalence can result from multiple factors beyond incidence decline, I think an epidemic modeling approach would have been more suitable from the start (whether individual-based or other). Such a model would have allowed for a more comprehensive assessment of all potential drivers of changing prevalence over time, and explore whether incidence has declined. Unfortunately, I think that the current approach does not fully answer the primary research question.

General comments

- The Abstract and Introduction highlights the lack of longitudinal studies among FSW in Zimbabwe/SSA and the fact that little is known about changes in incidence over time. But there is a major recent paper which looked at trends in HIV seroconversion over 2009-2019 among FSW in Zimbabwe, using programmatic data, which is completely omitted. This is cited later, in the Discussion but think it comes too late, and was surprised by this omission.
 - o Ref: Jones HS, Hensen B, Musemburi S, et al. Temporal trends in, and risk factors for, HIV seroconversion among female sex workers accessing Zimbabwe's national sex worker programme, 2009-19: a retrospective cohort analysis of routinely collected HIV testing data. *Lancet HIV*. 2023;10(7):e442-e52.
- The authors mention that surveys were done in 13 towns and 2 cities in Zimbabwe between 2013 and 2023. Please name them as it was difficult to understand whether there is overlap between surveys and where. Appendix 1 and 5 give some information on this, but having the names would help. What % of the FSW population in Zimbabwe is covered by these sites?
- Lines 411-413: "To account for the clustered nature of our data by site, we present cluster summary means and the minimum and maximum estimates (range) across all sites." Cluster summary means should be defined.
- Lines 421 -430: The authors indicate that they compared HIV prevalence across different surveys for the same birth cohorts, but it is not clear how the results were assessed: was it a visual comparison only for each age band? Were there any tests involved?
- I also did not follow which years are compared to examine trends in HIV prevalence: is it 2013 vs 2016 and vs 2021, and separately, 2017 vs 2023? Is it 2016 vs 2013 and 2021 vs 2013? Is it 2013 vs 2016-2017 vs 2021-2023? I found this key information to be inconsistently presented throughout the manuscript
 - o Related to the above, the following is unclear—what does it mean to test for HIV prevalence recently? This information

should be presented in the Methods: "To test whether HIV prevalence had declined recently we pooled data from all 15 sites for 2016-2017 (n=4,005) and 2021-2023 (n=4,475). Across all five surveys, 5,631 FSW tested HIV-negative and 4,931 tested HIV-positive." (lines 87-89)

(Remarks on code availability)

Reviewer #2

(Remarks to the Author)

I enjoyed reading this interesting paper which provides detailed information on HIV prevalence over time for a population that is very important for HIV epidemiology and for whom high quality data are typically sparse. The paper clearly describes the methods and analyses used to obtain these estimates and sets them in context of the epidemic trends in Zimbabwe.

The third paragraph of the results describes the estimates from the synthetic cohort approach. I think this is a reasonable approach in principle but have some concerns about this part of the analysis. Appendices 3 and 4 show that there is little power to estimate prevalence by single year of age, and the CI from some of the individual estimates presented in these tables run from 0 to 100%. It seems hard to conclude much based on such uncertain estimates. These data underlie Figure 1 d)-f) and although the trends are clearly downward at all ages in e) there is less pattern in the other two. Would it be better if the data were grouped by two or three years of age (depending on the interval elapsed between the two time points) so that there was a little more power for each estimate? I can see that it is nice to have the graphs on the right mirror the trend lines in the graphs on the left, and that grouping the trend estimates would detract from that, but it might reduce the noise evident in the yearly estimates. Some idea of the CIs would be useful on one or other set of plots.

A recurrent question, which first came to mind on line 176, is how the PrEP users compare to the non-users for condomless sex and STIs. The increase in condomless sex and STIs is about the same as the uptake of PrEP – is that just a coincidence?

Line 196: You mention increasing treatment among men as contributing to incidence decline, but the high levels of viral suppression in the study population are only mentioned in the appendix. To what extent are the FSW and client populations stable? Could suppression in these women be limiting transmission to their male partners to a large enough extent to also slow the flow of return infections from male partners?

In the discussion you note that entry and exit patterns into sex work may have changed post-pandemic, which may be why women reported knowing fewer FSW in the later period. Has the use of phones had an impact post-pandemic?

Methods line 399: You note that women with no HIV test were excluded but do not say how many that was in each round. It would be reassuring to see that, in case the proportion with no test was sufficiently different between the rounds to potentially explain some of the change in prevalence.

A few minor points:

Appendix 4: Age 20, the estimates are the same at both time points, is that correct?

Line 94: The last sentence in the second paragraph of the results doesn't read very well- the way the brackets are placed it isn't very clear. I'd suggest either removing the brackets and referring to the Table for the CI and ranges, or perhaps just reorganising the sentence to say it's a 15 percentage points drop from 38%.

Line 116: I think the comparison described in this sentence is 24 year olds in 2021 compared to 19 years olds in 2016.

Line 127: refers to Appendix 6 Figure 2 in the context of 18-39 year olds, but that graph is Figure 2 and I couldn't see a reference in the text to the age groups for which results are given in Appendix 6.

Table 1: In the headings for each study the total number of respondents is given as n and the number of people who tested positive in the row below is also given as n. Elsewhere N was used for the total which was clearer.

Figure 3: The legend says that circle 1 describes the 15 seed 1 participants, but I could only see 6.

Figures 5, 6, 9 and 10 would be easier to read, and compare, if they were presented in the same aspect ratio as Figs 3 and 4.

(Remarks on code availability)

Version 1:

Reviewer comments:

Reviewer #1

(Remarks to the Author)

Thank you to the authors for the detailed revisions and responses.

(Remarks on code availability)

Reviewer #2

(Remarks to the Author)

(Remarks on code availability)

I've looked at the code and it looks fine but I'm not an R user, and in any case you can't really judge without the data.

Reviewer #1 (Remarks to the Author):

The authors of this study aimed to examine trends in HIV prevalence among female sex workers (FSW) in Zimbabwe over time and to determine whether the anticipated decline in prevalence reflects a true decline in incidence. They analyzed data from five RDS surveys conducted between 2013 and 2023. To assess whether the observed decline in prevalence was due to declining incidence, the authors performed several exploratory analyses, including:

- Comparing HIV prevalence by birth cohort across surveys.
- Investigating potential biases in RDS recruitment over time.
- Examining changes in FSW demographics, risk behaviors, and engagement with HIV care.

The authors conclude that these analyses—collectively— suggest a decline in both HIV prevalence and incidence over time, a finding further supported by a parallel modeling analysis conducted by the authors.

I found the manuscript difficult to follow due to the lack of a clear central analysis and the presence of too many small, descriptive analyses. The methodology was not always well-explained—for instance, the trend analysis of HIV prevalence across different years and locations was difficult to interpret. Additionally, the simulation model used as comparison lacked sufficient methodological details to understand and evaluate and appeared out of place. Given the complexity of a model, a dedicated stand-alone paper with more details on model development and calibration would have been more appropriate.

We would like to thank the reviewer for their important comment seeking clarification on the main analysis of the manuscript. Our central analysis investigates whether there has been a significant decline in HIV prevalence over time. After establishing this decline, we perform a set of complementary analyses to explore various plausible explanations for the observed trend. We recognise that many factors, as you mentioned in your comment below, can influence shifts in HIV prevalence. Therefore, we conducted numerous analyses, in addition to the descriptive analyses, to investigate this further. We have now edited the manuscript, starting from the Study scope (lines 71-80) and the Methods to clarify that the main analysis is to assess whether the decline in HIV prevalence is real. We have also changed the order of the Methods and Results sections to focus on RDS data analysis first including all the checks we do before moving to triangulating the findings with the mathematical model.

We triangulated the analyses of empirical data with a well-established mathematical model that has been used extensively to guide HIV policy decisions. We acknowledge the complexity of this model and have included a detailed description of its structure in *Suppl Appendix B*. Additionally, in *Suppl Appendix B*, we have referenced multiple papers that describe its structure and fitting in more detail.

Lastly, since changes in prevalence can result from multiple factors beyond incidence decline, I think an epidemic modeling approach would have been more suitable from the start (whether individual-based or other). Such a model would have allowed for a more comprehensive assessment of all potential drivers of changing prevalence over time, and explore whether incidence has declined. Unfortunately, I think that the current approach does not fully answer the primary research question.

Thank you for your comment. We acknowledge that our assessment of the strength of evidence for declines in HIV prevalence was limited by the fact that we do not have mortality data and cannot quantify transitions into and out of sex work as well as migration. We have now more clearly acknowledged this and added these to the limitations in our Discussion section (lines 276-280). However, given the significant improvements in the HIV treatment cascade among FSW, as mentioned in lines 118-121, and the younger age group we restricted our analysis to, mortality is unlikely a major driver of the decline in HIV prevalence we observed. High proportions of viral suppression could rather have improved the survival rates among FSW living with HIV. Text in lines 276-280 reads: “Additionally, we were not able to assess the role of other drivers of HIV prevalence such as mortality, transitions into and out of sex work, and migration. Mortality might not have had a significant impact, as viral suppression has improved significantly over time among FSW, likely positively impacting survival rates. More research is needed to understand the potential impacts of population turnover resulting from these drivers on estimates of HIV incidence among FSW”

We agree that a mathematical model can be used to assess changes in prevalence by considering potential drivers that may not necessarily be accounted for using empirical data. However, we would like to stress here the uniqueness of the empirical data on which we base our primary analyses. Our serial cross-sectional surveys over 10 years using the same methodology are a unique resource in the region. Further, the model used, HIV Synthesis is a comprehensive model parameterised using data from these and other surveys. HIV Synthesis also takes into account other potential drivers of changes in prevalence, including the background epidemic and changes in risk behaviour, details of which have now been included in the *Suppl Appendix B*.

General comments

- **The Abstract and Introduction highlights the lack of longitudinal studies among FSW in Zimbabwe/SSA and the fact that little is known about changes in incidence over time. But there is a major recent paper which looked at trends in HIV seroconversion over 2009-2019 among FSW in Zimbabwe, using programmatic data, which is completely omitted. This is cited later, in the Discussion but think it comes too late, and was surprised by this omission.**

o Ref: Jones HS, Hensen B, Musemburi S, Chinyanganya L, Takaruza A, Chabata ST et al. Temporal trends in, and risk factors for, HIV seroconversion among female sex workers accessing Zimbabwe's national sex worker programme, 2009-19: a retrospective cohort analysis of routinely collected HIV testing data. *Lancet HIV*. 2023;10(7):e442-e52.

Thank you. This work was undertaken by our group. We have now cited the paper both earlier and later in the manuscript (lines 37 and 254). We note that the papers use different methods and data sources. The Lancet HIV paper is based on FSW who attend a targeted FSW programme and will not capture those who do not. In contrast, the RDS samples we analyse here include both FSW who have and have not attended the programme and will be more representative. The papers have different strengths and limitations, which we discuss in both this and the cited paper. The fact that both point to a decline in HIV seroconversion/incidence rates adds strength to our overall interpretation.

- **The authors mention that surveys were done in 13 towns and 2 cities in Zimbabwe between 2013 and 2023. Please name them as it was difficult to understand whether there is overlap between surveys and where. Appendix 1 and 5 give some information on this, but having the names would help. What % of the FSW population in Zimbabwe is covered by these sites?**

Thank you. We think it's clear in lines 86-90 that there was overlap and say that we selected these sites because they were the same throughout: "We analysed data for 6,240 FSW aged 18-39 recruited from the same 13 locations in 2013, 2016 and 2021, and for 4,322 FSW recruited from the same two cities in 2017 and 2023". However, we would welcome editorial advice on the naming of sites. Our own view is that naming the places poses some (low probability) risks to participants and to the communities in question, without any major scientific benefits. Over the 15 years we have worked with the national programme for sex workers in Zimbabwe, we have generally avoided naming specific, particularly smaller locations in papers, with some exceptions. This remains our preferred approach. Instead, we have included the site codes in *Suppl Appendix A, Table 1*, making it clearer where the overlaps are. We are happy to discuss further if the advice is to include the names.

From the national population size estimation exercise we conducted in 2024 (data yet to be published), we estimated that these 15 locations include approximately 45% of the total FSW population in Zimbabwe. We have now included this information in the paper (lines 410-411).

• **Lines 411-413: "To account for the clustered nature of our data by site, we present cluster summary means and the minimum and maximum estimates (range) across all sites." Cluster summary means should be defined.**

Thank you – we have now defined cluster summary means in lines 429-432, and it reads, "To account for the clustered nature of our data by site, we calculate prevalence estimates for each site (cluster), sum them, and divide by the number of sites (n=13 for 2016-2017 pooled data and n=2 for 2021-2023 pooled data)." We now refer to these estimates as "cluster prevalence means"

• **Lines 421 -430: The authors indicate that they compared HIV prevalence across different surveys for the same birth cohorts, but it is not clear how the results were assessed: was it a visual comparison only for each age band? Were there any tests involved?**

Thank you for your question. We did not formally compare HIV prevalence across different survey rounds for the same birth (age) cohorts, and we do not base our conclusions of declining HIV prevalence on single year comparisons of age cohorts – the graphs are primarily for visualisation purposes. In the text, we present descriptive statistics of the mean changes in HIV prevalence within the age cohorts, without conducting a formal comparison. Comparisons were done on pooled age-specific (not age cohorts) HIV prevalence using a t-test and we present the 95% CI and the p-value (*Suppl Appendix A, Tables S4*). We have now made this clear the paragraph in lines 444-453.

• **I also did not follow which years are compared to examine trends in HIV prevalence: is it 2013 vs 2016 and vs 2021, and separately, 2017 vs 2023? Is it 2016 vs 2013 and 2021 vs 2013? Is it 2013 vs 2016-2017 vs 2021-2023? I found this key information to be inconsistently presented throughout the manuscript**

Thank you. We have now made edits to the Methods section to make it clear which years we compared for trends in HIV prevalence. We compared 2013 vs 2016, 2016 vs 2021, and 2017 vs 2023. The paragraph now reads "... For each survey we calculated cluster mean RDS-weighted age-specific HIV prevalence and plotted these for the same 13 sites for years 2013 and 2016, and for 2016 and 2021, as well as for 2 different sites for 2017 and 2023. We also present age-specific HIV prevalence for pooled survey rounds which each include the same 13+2 sites (2016-2017 and 2021-2023) and calculated 95% CI and p-value for the difference in means...." (lines 437-441). We have also made

edits to other sections of the Methods to make sure that it's clear where we compare individual survey rounds (2013, 2016, 2017, 2021, 2023) data, and pooled 2016-2017 and 2021-2023 data (lines 355-363 and lines 410-434).

o Related to the above, the following is unclear—what does it mean to test for HIV prevalence recently? This information should be presented in the Methods: "To test whether HIV prevalence had declined recently we pooled data from all 15 sites for 2016-2017 (n=4,005) and 2021-2023 (n=4,475). Across all five surveys, 5,631 FSW tested HIV-negative and 4,931 tested HIV-positive." (lines 87-89)

Thank you. We realise the use of the word recently is confusing and we have now taken it out and revised the sentence to "To test whether HIV prevalence had declined between 2016-2017 and 2021-2023, we pooled and calculated cluster prevalence mean using data from all 15 sites for 2016-2017 (n=4,005) and 2021-2023 (n=4,475), excluding data from 2013 (n=2,082) (lines 88-90).

Reviewer #2 (Remarks to the Author):

I enjoyed reading this interesting paper which provides detailed information on HIV prevalence over time for a population that is very important for HIV epidemiology and for whom high quality data are typically sparse. The paper clearly describes the methods and analyses used to obtain these estimates and sets them in context of the epidemic trends in Zimbabwe.

We appreciate the positive comments above.

The third paragraph of the results describes the estimates from the synthetic cohort approach. I think this is a reasonable approach in principle but have some concerns about this part of the analysis. Appendices 3 and 4 show that there is little power to estimate prevalence by single year of age, and the CI from some of the individual estimates presented in these tables run from 0 to 100%. It seems hard to conclude much based on such uncertain estimates. These data underlie Figure 1 d)-f) and although the trends are clearly downward at all ages in e) there is less pattern in the other two. Would it be better if the data were grouped by two or three years of age (depending on the interval elapsed between the two time points) so that there was a little more power for each estimate? I can see that it is nice to have the graphs on the right mirror the trend lines in the graphs on the left, and that grouping the trend estimates would detract from that, but it might reduce the noise evident in the yearly estimates. Some idea of the CIs would be useful on one or other set of plots.

Thank you. We appreciate that grouping age cohort data into two or three age bands would increase the power. However, we believe that we have sufficient power to estimate prevalence by single year, as demonstrated by the n/N data columns now included as *Suppl Appendix A, Tables S5*. Additionally, we do not base our conclusions of declining HIV prevalence on single year (of age) comparisons in age cohorts – this is data underlying Figure 1d-f and is primarily for visualisation purposes. Also, in *Suppl Appendix A, Tables S3* (age-specific HIV prevalence by survey round – data underlying Figure 1a-c) and *S4* (pooled age-specific HIV prevalence), we present the cluster means and ranges (with ranges running from 0-100, not due to lack of power), and the 95% CI are only presented in *Suppl Appendix A, Tables S4* where data is pooled. Given that we are presenting the ranges, we do not think it is useful to include them in the plots.

A recurrent question, which first came to mind on line 176, is how the PrEP users compare to the non-users for condomless sex and STIs. The increase in condomless sex and STIs is about the same as the uptake of PrEP – is that just a coincidence?

In the 2021-2023 data, where we have information on PrEP use, the prevalence of PrEP use was very similar among FSW who reported condomless sex and those who tested positive for Gonorrhoea, Chlamydia, or Trichomoniasis (26.3% vs 25.6%). PrEP data were not available for the earlier years, so we were unable to assess changes in PrEP use over time. The similarity between the increase in condomless sex and STIs and the uptake of PrEP appears to be coincidental.

Line 196: You mention increasing treatment among men as contributing to incidence decline, but the high levels of viral suppression in the study population are only mentioned in the appendix. To what extent are the FSW and client populations stable? Could suppression in these women be limiting transmission to their male partners to a large enough extent to also slow the flow of return infections from male partners?

Thank you. We agree that this is potentially part of the explanation, although it is challenging to assess with the available data. We have included comprehensive viral suppression data (*Suppl Appendix A, Tables S9-S10*) to demonstrate the significant improvements over time. This improvement potentially reduces the risk of transmission to male partners, aligning with the treatment as prevention strategy, and may decrease the likelihood of infections of HIV-negative female sex workers from male partners. We have added a discussion on this point in the revised manuscript (lines 324-326) – “Additionally, the significant improvements of viral suppression among FSW over time, may have potentially reduced the risk of transmission to HIV-negative male partners, which in turn may have further decreased the likelihood of HIV-negative FSW becoming infected by male partners”

In the discussion you note that entry and exit patterns into sex work may have changed post-pandemic, which may be why women reported knowing fewer FSW in the later period. Has the use of phones had an impact post-pandemic?

Thank you for an interesting question. We agree that use of phones could also have had an impact on why women reported knowing fewer FSW post-pandemic, but we don't have data to support that. Therefore, we have included it in the discussion (lines 289-292) – “Interestingly, women in our study reported knowing fewer other sex workers in later survey rounds. This may indicate changes in sex worker networks, possibly due to shifts in patterns of entry and exit from sex work or a transition to phone or internet-based sex work post-COVID pandemic”

Methods line 399: You note that women with no HIV test were excluded but do not say how many that was in each round. It would be reassuring to see that, in case the proportion with no test was sufficiently different between the rounds to potentially explain some of the change in prevalence.

Thank you. Very few women (16 in 2013, 0 in 2016, 19 in 2017, 1 in 2021, and 0 in 2023) had no HIV test and were excluded from the analysis (*Suppl Appendix A, Table S1*). This is insignificant in explaining the change in HIV prevalence. We have now highlighted in text the number of women who were excluded from the analysis because of missing HIV test (lines 82-86).

A few minor points:

Appendix 4: Age 20, the estimates are the same at both time points, is that correct?

Thank you. We have checked data for Appendix 4, now *Suppl Appendix A, Table S1* and yes, the estimates for age 20 are identical at both time points and the ranges are also identical.

Line 94: The last sentence in the second paragraph of the results doesn't read very well- the way the brackets are placed it isn't very clear. I'd suggest either removing the brackets and referring to the Table for the CI and ranges, or perhaps just reorganising the sentence to say it's a 15 percentage points drop from 38%.

Thank you. We have reorganised the sentence to make it clearer "In the 15-site pooled analysis, cluster mean RDS-weighted HIV prevalence in 2021-2023 was 38.9% (cluster range: 22.6–50.9), which was a 15.7% drop from 54.6% (cluster range: 36.2–72.3) in 2016-2017 (95% CI: 9.2–22.2; p-value <0.001) (Table 1) – relative risk = 0.74 (95% CI: 0.68-0.79) (*Suppl Appendix A, Table S9*)." (lines 97-100).

Line 116: I think the comparison described in this sentence is 24 year olds in 2021 compared to 19 years olds in 2016.

Thank you. We have corrected the wording to make the sentence clear. "The main exception to this pattern was among young FSW in 13 of the sites where HIV prevalence increased by 8.5% among 24 year olds in 2021 compared to 19 year olds in 2016, and by 8.3% among 25 year olds in 2021 compared to 20 year olds in 2016." (lines 121-124).

Line 127: refers to Appendix 6 Figure 2 in the context of 18-39 year olds, but that graph is Figure 2 and I couldn't see a reference in the text to the age groups for which results are given in Appendix 6.

Thank you. In the text, we want to report data for 18-39 year olds without further breaking it down into 18-24 and 25-39, as shown in *Suppl Appendix A, Figure S12-S13* (formerly Appendix 6). We have therefore now removed a reference to the Appendix and the sentence now reads "HIV incidence was lower when modelled in the context of a FSW programme, from 2010 onwards, declining from 7.7 (0.0-32.2) per 100py in 2016 to 4.1 (0.0-12.8) per 100py in 2021 (Figure 3b)." (lines 190-192).

Table 1: In the headings for each study the total number of respondents is given as n and the number of people who tested positive in the row below is also given as n. Elsewhere N was used for the total which was clearer.

Thank you for spotting that. We have now corrected it to ensure that N represents the total number of respondents, and n represents the number of respondents who tested positive throughout Table 1.

Figure 3: The legend says that circle 1 describes the 15 seed 1 participants, but I could only see 6.

Thank you. We realise it's not clear in the legend, but we combined all seeds 1 from the 15 sites to circle 1, all seeds 2 from the 15 sites to circle 2, all seeds 3 from the 15 sites to circle 3, and all other the seed numbers up to seed 6. We have clarified that in the legend of now Figure 2. The legend now reads as below:

Figure 2: Combined RDS recruitment trees for 15 pooled survey sites for the period 2016-2017 and 2021-2023

Circle 1, for example, depicts the 15 participants who were seed 1 across the 15 sites surveyed in 2016-17. Circle 2 depicts a total of 15 participants who were seed 2 across the 15 sites, and so on up to circle 6, which depicts a total of 15 participants who were seeds 6 across the 15 sites. The circles connected to these circles 1, 2, 3, 4, 5 and 6, depict participants who were recruited by seeds 1 across the 15 sites and this goes on and on until the final wave 7. The colour coding for HIV prevalence indicates the prevalence in each wave, with everyone in each wave being assigned the same colour.

Figures 5, 6, 9 and 10 would be easier to read, and compare, if they were presented in the same aspect ratio as Figs 3 and 4.

Thank you. We have changed the aspect ratio for all Figures in Appendix 6 (now *Suppl Appendix A, Figure S1-S10*) to match that of Figures 3 and 4.